# Evaluation of Immunogenicity of an Orf Virus Vector-Based Vaccine Delivery Platform in Sheep

**DOI:** 10.3390/vaccines13060631

**Published:** 2025-06-11

**Authors:** Sean R. Wattegedera, Jackie Thomson, Lesley Coulter, Ann Wood, Rebecca K. McLean, Holly Hill, Cameron Cunnea, Karen Snedden, Ann Percival, Javier Palarea-Albaladejo, Gary Entrican, David Longbottom, David J. Griffiths, Colin J. McInnes

**Affiliations:** 1Moredun Research Institute, Pentlands Science Park, Penicuik EH26 0PZ, UKrebecca.mclean@pirbright.ac.uk (R.K.M.); holly.hill@moredun.ac.uk (H.H.); cameron.cunnea@moredun.ac.uk (C.C.); karen.snedden@moredun.ac.uk (K.S.); gary.entrican@roslin.ed.ac.uk (G.E.); david.longbottom@moredun.ac.uk (D.L.); david.griffiths@moredun.ac.uk (D.J.G.); colin.mcinnes@moredun.ac.uk (C.J.M.); 2Biomathematics and Statistics Scotland, Edinburgh EH9 3FD, UK; javier.palarea@udg.edu

**Keywords:** virus-based vaccine, immunogenicity, platform vaccine development, major outer membrane protein, *Chlamydia abortus*, humoral and cellular immunological analysis

## Abstract

Background/Objective: Virus-based vaccine vectors have been widely utilised in commercial vaccines, predominantly for virus infections. They also offer promise for bacterial diseases, for which many vaccines are sub-optimal or ineffective. It is well-established for chlamydial infections, including ovine enzootic abortion, that the major outer membrane protein (MOMP) antigen is protective. Immune responses strongly associated with controlling *Chlamydiae* include cellular interferon-gamma (IFN-γ) production. Methods: A study was conducted to compare the ability of a modified Orf virus vector directly with a modified sheep maedi visna virus vector to deliver the *C. abortus* antigen *ompA* and stimulate vaccine-induced responses in sheep. The Orf virus-based vaccine (mORFV-*ompA*) was found to be more effective in stimulating MOMP-specific antibodies and cellular antigen-driven IFN-γ in immunised sheep. This mORFV-*ompA* vaccine was assessed in a follow-up immunogenicity investigation in sheep, where the cellular and humoral immune responses elicited following immunisation with the live or inactivated vaccine were determined. Sheep were immunised intramuscularly with a live mORFV-*ompA* (*n* = 10) or an inactivated mORFV-*ompA* (*n* = 10). An additional group of 10 sheep served as unvaccinated controls. Results: Serological anti-MOMP antibodies and cellular recall responses of peripheral blood mononuclear cells to the native *C. abortus* antigen were assessed. Immunisation with either the live or inactivated mORFV-*ompA*-induced anti-MOMP immunoglobulin-G. Antigen-specific cellular responses, characterised by the secretion of IFN-γ and interleukin (IL)-17A, with negligible IL-10 and no IL-4, were detected in lymphocyte stimulation assays from both mORFV groups. No antibody responses to the mORFV platform were detected following immunisations. Conclusions: Both live and inactivated vaccines have the potential to be a platform technology for deployment in sheep. This addresses a notable gap in veterinary vaccine development where the induction of both humoral responses and cellular responses is required without using an adjuvant. The successful use of the MOMP candidate antigen suggests potential utility for bacterial disease deployment.

## 1. Introduction

Recombinant virus vector technology is an established vaccine “plug-and-play” platform that has successfully been used to develop a new generation of vaccines that utilise a subunit antigen over a more conventional whole pathogen approach [1]. These recombinant vaccines offer many of the best features of whole pathogen vaccines, including appropriate stimulation of innate and adaptive immune responses (including antibody and cellular responses) and benefit from cost-effective and reliable in vitro production at scale as demonstrated by commercial vaccines to COVID-19 during the recent pandemic [2,3]. The safety of recombinant vector vaccines has previously been identified as a potential barrier to deployment for both humans and animals [4,5]. However, a number of poxvirus-based vaccines have already been commercialised for viral diseases of animals, including Newcastle Disease virus (Trovac^®^-NDV, Vectormune^®^ ND), Fowlpox virus and Infectious Laryngotracheitis (Vectormune FP LT), and avian influenza (Trovac^®^ Al H5 and Trovac Prime H7) for chickens, as reviewed by Wang et al. [6].

There is a notable gap in commercial vaccines for bacterial diseases adopting the virus vector technology [7]. To address this, we have used data from existing commercial bacterial vaccines (live, live-attenuated, and whole pathogen-inactivated preparations) to select a candidate protective antigen for deployment in virus vector platforms. This antigen is the major outer membrane protein (MOMP) present in Gram-negative bacteria, including chlamydial species [8]. There is a large body of evidence across human, farm animal and wildlife host species on the protective efficacy of this antigen [9,10,11]. We have selected the MOMP gene *ompA* from *Chlamydia abortus*, a major abortifacient pathogen of sheep, for incorporation into two virus vector systems we have developed. They are based on two viruses of sheep endemic to the UK, a sheep parapoxvirus Orf and a sheep lentivirus, which can be engineered for safely delivering potentially protective antigens.

Orf Virus has a large (~135 kbp) linear double-stranded DNA genome containing both genes that are essential to virus replication and those that are non-essential to replication but confer an advantage to the virus in the face of the host’s immune response. The engineering of poxviruses into vaccines has focussed on the manipulation of regions of the genome less critical for virus fitness for insertion of foreign antigen(s) [12,13]. These vaccines have the ability to stimulate both antigen-specific humoral and cellular immunity [14]. Replication of the live poxvirus vaccines occurs in the cytosol of the recipient host cells and, therefore, reduces the opportunity to recombine with the DNA of the host organism [6,15]. To date, most studies using Orf as a vaccine vector have been conducted on non-permissive hosts, including mice, cats, and dogs. Relatively little data is known about vaccination in permissive hosts such as sheep. Using the sheep parapoxvirus Orf (ORFV), we have developed a modified ORFV-vectored vaccine (mORFV) with the NZ2 strain [12] to develop a new prototype vaccine against OEA.

In comparison to poxviruses, lentiviruses are less well-established as vaccine vectors, but several studies have shown that they are potent stimulators of protective immunity for vaccine targets [16,17], including viruses (e.g., West Nile virus, Japanese encephalitis virus, and influenza virus) [18,19,20,21] and protozoa (e.g., malaria and Naegleria fowleri) [22,23]. As with poxviral vectors, lentiviral vectors stimulate both cellular and humoral immune responses to the expressed transgene. Lentiviruses offer other benefits as vaccine vectors, including their capacity for stable gene delivery to non-dividing cells, including antigen-presenting cells. Secondly, lentiviral-mediated vaccine delivery does not result in the expression of any viral genes within the targeted animal; hence, there is negligible anti-vector immunity. Since HIV-based lentiviral vectors do not efficiently infect sheep antigen-presenting cells [24], we have developed a lentiviral vector system from the ovine lentivirus, maedivisna virus (mMVV), which is able to transduce ovine macrophages and dendritic cells much more efficiently than vectors derived from HIV-1 in vitro [25].

In the prototype vaccines, we have utilized the *C. abortus* gene *ompA* expressing the dominant immunogenic MOMP of the efficacious subcellular chlamydial outer membrane complex (COMC) prototype vaccine [26,27]. In a primary investigation, we compared two groups of sheep immunised with the *ompA* gene using the two vector systems, mORFV and mMVV, assessing their ability to stimulate MOMP-specific antibody and antigen-driven interferon-gamma (IFN-γ) responses. Based on the results observed, a follow-up study was conducted using the more promising platform technology to comprehensively assess its immunogenic capacity as a live or inactivated vaccine.

## 2. Materials and Methods

### 2.1. Project Overview

This project comprised two immunogenicity studies to identify and evaluate which of two virus vectors, mORFV and mMVV, which were engineered to express the recombinant protein product (MOMP) of the *ompA* gene, induced the most appropriate immune responses in sheep (Figure 1).

### 2.2. Production and Evaluation of the Recombinant Modified Orf Virus Encoding the ompA Gene of C. abortus (mORFV-ompA)

The mORFV-*ompA* virus was generated by homologous recombination using the well-characterised ORFV NZ2 virus strain (Genbank accession no. DQ184476) [12] that we had previously modified to express the green fluorescent protein (GFP) (strain mORFV-*gfp*; unpublished) and a plasmid construct containing the *C. abortus ompA* gene (GenBank accession CR848038, gene CAB048) (Figure 2), which encodes the 40 kDa MOMP. mORFV-*gfp* was generated by replacing gene 117 of ORFV NZ2 with the *gfp* gene and the neomycin resistance and β-galactosidase genes: gene 117 encodes a non-essential protein that has granulocyte macrophage-colony stimulating factor (GM-CSF) and interleukin (IL)-2 binding properties [28]. The inclusion of the *gfp* gene enabled differentiation of the ‘clear’ plaques produced by the *ompA*-modified virus from the ‘green’ plaques produced by the parent virus vector. The *ompA* plasmid (pSP72-*ompA*, Figure 2) was constructed by cloning the left viral flanking region of ORFV (1499 bp; genes 115 and 116), an ORFV early/late promoter (42 bp), the *ompA* gene (1170 bp) and the right viral flanking region of ORFV (1794 bp, comprising 150 bp from the 3′ end of gene 117 and complete genes 118 and 119) into commercial cloning vector pSP72 (Promega, Southampton, UK, #P2191) by PCR, targeting appropriate restriction enzyme sites and using the cloning protocols outlined in the pSP72 Vector Technical Bulletin (Promega) and by Krieg and Melton [29]. pSP72-*ompA* was produced for use in transfections by transforming JM109 cells (Promega, #L2005) and culturing in successive rounds of L-broth culture at 37 °C using protocols described in the *E. coli* Competent Cells Technical Bulletin (Promega) and by Schmidt [30] and an endonuclease-free stock of plasmid DNA was prepared using an Endo-free Plasmid Maxiprep Kit (Qiagen, Crawley, UK; #12362).

To generate mORFV-*ompA*, 1 µg pSP72-*ompA* was transfected into foetal lamb skin (FLS) primary cells in six-well plates using a Cell Line Nucleofector Kit L^®^, Program T-030 (#VCA-1005, Lonza, Basel, Switzerland). After incubating for 3 h at 37 °C, the transfected cells were infected with parental virus (mORFV-*gfp*) at a multiplicity of infection (MOI) of 1 in Medium 199 (Merck, Dorset, UK, #M0650) supplemented with 2% heat-inactivated (ΔH) foetal bovine serum (FBS, Merck Life Science UK Ltd. (Sigma-Aldrich Co.), Gillingham, UK; #F2442, USA origin), 2 mM l-glutamine (Merck, #49419), 0.15% sodium hydrogen carbonate (Merck, #6014) and 10% tryptose phosphate broth (Merck, #T8159) and incubated at 37 °C/5% CO_2_ until plaques were observed (approximately 4 days). ‘Clear’ plaques produced by the mORFV-*ompA* recombinant virus, following the homologous recombination of mORFV-*gfp* and the transfected pSP72-*ompA* (Figure 2), were isolated from the parental ‘green’ plaques by successive rounds of plaque purification. The determination of ‘clear’ plaques was assessed by light microscopy and confirmed by specific PCR. Here, FLS cells infected with mORFV-*ompA* were harvested at 3 days and 6 days post-infection. Cells were lysed into RLT solution (Qiagen, #74104) containing beta-mercaptoethanol (Merck, #M6250), and RNA was prepared using an RNA prep kit with Qiashredder (Qiagen, #74104). A one-step RT-PCR and combined gDNA digestion was performed using the Superscript IV One-Step RT-PCR System and EZDNase (Invitrogen, Thermo Fisher Scientific, Leicestershire, UK, #12595025) to amplify *ompA* fragment (Appendix A).

Viral DNA was isolated and sequenced using various sequencing primers corresponding to genes 116 and 118 of ORFV and the *ompA* gene to confirm that the recombined sequence was correct. A single viral plaque was then amplified to create virus stocks for downstream studies. The mORFV-*ompA* virus was titrated on FLS cells using twelve replicates from 10^−2^ to 10^−8^ alongside no virus control wells in tissue culture flatbottom plates (Thermo Fisher Scientific, #07-200-90) and allowed to incubate at 37 °C for 7–10 days for the determination of TCID_50_.

### 2.3. Production of the Recombinant Modified Maedi Visna Virus Encoding the ompA Gene of C. abortus (mMVV-ompA)

mMVV vaccine vectors utilised a four-plasmid MVV-derived lentiviral vector system comprising an MVV gene transfer vector, an MVV Gag-Pol packaging construct, and plasmids expressing MVV Rev and vesicular stomatitis glycoprotein G (VSV-G) [25,31]. To construct the MVV transfer vector expressing *C. abortus* MOMP, *ompA* was isolated by PCR from *C. abortus* DNA and inserted downstream of the human cytomegalovirus (CMV) immediate-early promoter (Figure 3). An N-terminal signal peptide from the *Ovis aries* immunoglobulin kappa-2.1 light chain was added to mediate membrane targeting in eukaryotic cells. A C-terminal FLAG epitope tag was added to aid detection of protein expression.

The mMVV-*ompA* vector particles were produced by co-transfection of human embryonic kidney 293T cells with the four vector plasmids (Figure 3) using FuGENE-HD transfection reagent (Promega, #E2311), according to the manufacturer’s guidelines. 293T cells were cultured in Iscove’s modified Dulbecco’s Medium (Merck, #I3390), supplemented with 4 mM glutamine and 10% ΔH FBS. Briefly, 293T cells were plated in T75 tissue culture flasks to achieve approximately 80% confluency on the day of transfection. Transfection complexes were prepared by combining 3.6 μg MVV packaging construct (pCAG-MV-GagPol-CTE2X), 5.4 μg MVV transfer vector encoding *ompA* (pCVW-CMV-*ompA*), 1.2 μg MVV Rev expression construct (pCMV-Rev), and 1.8 μg VSV-G expression construct (pMD2.G: Addgene plasmid #12259; a gift from Didier Trono, EPFL, Lausanne, Switzerland), diluted in 1200 µL of serum-free medium (OptiMEM; Gibco, Fisher Scientific, Loughborough, UK; #51985026) and 36 µL of FuGENE-HD. Following incubation at RT for 15 min, FuGENE-DNA complexes were added to cells in 7 mL of fresh medium per flask, and cells were incubated at 37 °C for 16–18 h before the medium was replaced with 10 mL fresh IMDM supplemented with 10% FBS. Cell culture supernatant containing vector particles was harvested 42–48 h post-transfection and centrifuged at 300× *g* for 5 min before filtration through a 0.45 μm cellulose acetate filter (Sartorius, Gillingham, UK, #11106-100------G). Viral particles were pelleted at 70,000× *g* for 2 h at 4 °C in an Optima L-90K ultracentrifuge using an SW32 Ti rotor (Beckman Coulter, Buckinghamshire, UK) and resuspended at 100× concentration in phosphate-buffered saline (PBS), and stored frozen in aliquots at −70 °C.

### 2.4. Evaluation of mMVV-ompA

#### 2.4.1. Detection of Protein Expression

To confirm the expression of MOMP protein from the mMVV-*ompA* vector, the permissive cell line Crandell-Rees Feline Kidney (CRFK) (cultured in Dulbecco’s modified Earle’s medium supplemented with 2 mM glutamine, 1% non-essential amino acids (Gibco) and 10% FBS) was transduced with mMVV-*ompA* and cell lysates prepared 72 h post-transduction. Cell lysates were analysed using SDS-PAGE and immunoblotting. Blots were probed with a monoclonal antibody (mAb) to the FLAG epitope (mouse anti-FLAG M2 mAb, Merck, #F1804, diluted 1:1000) and rabbit anti-mouse HRP-conjugated secondary antibody (Dako, Agilent Technologies UK Ltd., Cheshire, UK, #P0266, 1:1000)—shown in Appendix A.

#### 2.4.2. Determination of mMVV-*ompA* Vector Titre

The infectious titre of vector mMVV-*ompA* was determined by TCID_50_ assay. Briefly, CRFK cells were plated in 96-well plates (1 × 10^4^ cells per well, Thermo Fisher Scientific, #07-200-90) and cultured overnight. Ten-fold serial dilutions (10^−1^–10^−8^) of mMVV-*ompA* were then plated onto these cells (8 wells per dilution) in a final volume of 200 µL per well with 4 μg/mL Polybrene (Merck, #TR-1003) and cultured for 72 h. The medium was removed, and cells were washed once in PBS before fixing in ice-cold methanol/acetone (1:1; 200 µL/well) for 15 min at RT and washed twice with PBS (200 µL/well). Cells were blocked with PBS/5% FBS for 1 h, and MOMP protein detected by immunocytochemistry using a primary antibody to the FLAG epitope tag (Merck #F1804, diluted 1:500) and secondary goat-anti-mouse IgG conjugated to beta-galactosidase (Southern Biotechnology, 2B Scientific, Bicester, UK; #1030-06 diluted 1:200). Beta-galactosidase activity was detected as described previously [32]. Positive wells were scored as containing one or more positive cell foci, and TCID_50_ was calculated using the Reed–Muench method [33].

### 2.5. Experimental Animals and Ethics Statement

The Moredun Animal Welfare and Ethical Review Body approved the experimental protocols described for both of the in vivo vaccine immunogenicity studies (Study A, permit number E52/18, approved 25 October 2018; and Study B, E48/19, approved 22 October 2019). All of the husbandry practices, welfare checks and animal procedures were carried out in strict accordance with the Animals (Scientific Procedures) Act 1986, as well as in compliance with all UK Home Office Inspectorate regulations and ARRIVE guidelines 2.0 [34]. Sheep were monitored at least twice daily throughout the duration of the study for any clinical signs or welfare issues.

### 2.6. Selection of Sheep for the Immunogenicity Studies

Sheep were obtained from the Moredun Research Institute breeding flock for Study A (6-month-old Texel-cross Scottish Mules: *n* = 30) or from disease-free OEA-accredited high-health flocks (participating in Scotland’s Rural College Premium Sheep and Goat Health Schemes [35]; 18-month-old Scottish Mules; *n* = 50) for Study B. In both cases, animals were pre-screened to ensure their OEA disease-free status and to ensure they were negative for other abortifacient agents, including pestiviruses, bovine viral diarrhoea virus (types 1 and 2) and border disease virus. For each animal, two 10 mL diagnostic blood samples were taken for serological analysis into BD Vacutainer^®^ serum tubes (Fisher Scientific; #12957686) and BD Vacutainer^®^ heparin tubes (Fisher Scientific; #13171543) for peripheral blood mononuclear cell (PBMC) stimulation/recall assays. The serum was screened by rOMP90-3 enzyme-linked immunosorbent assay (ELISA), as previously described [36]. Animals were excluded from the study if they had test values normalised to positive and negative controls above 60%. In addition, cellular immune recall and stimulation assays were conducted on PBMC isolated from whole blood using standard protocols, as previously revised [37]. Animals were excluded on the basis of low IFN-γ responses to the T cell mitogen Concanavalin A (ConA at 5 µg/mL, from *Canavalia ensiformis,* Merck, #C0412) and high IFN-γ responses to COMC antigen (0.5 µg/mL) and *C. abortus* elementary body (EB) antigen (1 µg/mL) for two cellular pre-vaccination sample points. A final group of twelve sheep was selected for study A (three groups of four sheep) and 30 sheep for study B (three groups of ten). Groups were balanced for their capacity to respond to ConA, as undertaken previously for a *C. abortus* pathogenesis study [37].

### 2.7. Preparation and Administration of the ompA Virus Vaccines and Blood Sample Schedules for Immunogenicity Studies A and B

For Study A, cell-free supernatants containing mORFV-*ompA* (10^7^ TCID_50_/mL) or mMVV-*ompA* (10^6^ TCID_50_/mL) were prepared in a volume of 1 mL for both primary and booster immunisations. The vaccines were delivered intramuscularly to two groups of four sheep (group 1, mORFV-*ompA*; group 2, mMVV-*ompA*) using a 19G 1” needle using a luer lock syringe to the right hind leg muscle over the prefemoral lymph node. This constituted day zero of the immunogenicity Study A (Figure 4). Four sheep served as non-immunised negative controls (group 3). Booster immunisations (BI) were delivered 38 days later. Injection sites were monitored on days 1, 3, and 7 post-primary immunisation (PI) for signs of any lesions.

The most promising virus vector from Study A was chosen for a more comprehensive assessment of immunogenicity in a larger sheep study (Study B), following delivery as either a live or inactivated vaccine. In order to derive the inactivated vaccine, the chosen live vaccine stock was divided, diluted to 10^7^ TCID_50_/mL and then inactivated by treatment with 0.05% *v*/*v* β-propiolactone (Thermo Fisher Scientific, #B23197) for 16 h at 4 °C. Following inactivation of the vaccine virus, the β-propiolactone was itself inactivated by incubation at 37 °C for 2 h. The β-propiolactone -treated virus was assessed alongside the untreated live vaccine by three successive rounds of blind passaging in susceptible FLS cells to check for cytopathic effects (CPE). The β-propiolactone treated virus was given a second round of treatment and confirmed free of CPE in susceptible cells for three successive rounds of sub-culture. Cell lysates were collected and analysed by RT-PCR as described in Section 2.2 to determine *ompA* transcription.

The vaccines were prepared as undertaken for Study A with identified *ompA*-vaccine (10^7^ TCID_50_/mL or 10^6^ TCID_50_/mL) equivalent in a volume of 1 mL for both primary and booster immunisations. The vaccines were delivered intramuscularly to two groups of ten sheep (group 1, inactivated vaccine; group 2, live vaccine) using a 19G 1” needle using a luer-lock syringe to the right hind leg muscle over the prefemoral lymph node (Figure 4). Ten sheep served as non-immunised negative controls (group 3). Booster immunisations were delivered 38 days later.

For both studies, the primary immunisations were delivered on designated day zero. Blood samples were collected throughout both studies for humoral and cellular analyses, with some differences in timing (Figure 4).

### 2.8. Serological Responses to C. abortus MOMP

Serum was collected from whole blood throughout both studies at the sample points highlighted in Figure 4. Samples were analysed for the presence of antibodies to *C. abortus* MOMP using an ID Screen^®^ *Chlamydophila abortus* Indirect Multi-Species ELISA kit (Innovative Diagnostics Vet, Grabels, France, #CHLMS-MS-5P), following the manufacturer’s instructions. Serum samples from a challenge control post-abortion sheep from a previous experimental study served as additional positive control samples in the ELISAs.

### 2.9. Cellular Analysis

PBMC isolated at the sample points outlined in Figure 4 were used for antigen-recall assays, as previously described [37], where cells were adjusted to 2 × 10^6^ cells/mL in IMDM supplemented with 10% ΔH FBS, 2 mM l-glutamine, gentamicin (50 µg/mL), 100 IU/mL penicillin, 50 μg/mL streptomycin and 50 µM beta-mercaptoethanol (all Merck, #59202C, #G1522, #P4458 and #M6250). 100 µL of PBMC were added to U-bottom ELISA plates (Fisher Scientific, Loughborough, UK, #TKT-180-050D) with 100 µL of *C. abortus* EBs (1 µg/mL) or COMC (0.5 µg/mL) in lymphocyte stimulation assays (LSAs) for 96 h in a humidified incubator (ThermoScientific, Heracell™ 150i CO_2_ Incubator) at 37 °C, 5% CO_2_. The T cell mitogen control, Con A (5 µg/mL) and medium alone were also included as positive and negative controls, respectively. Culture supernatants from quadruplicate wells for each stimulation condition and animal were pooled and stored at −20 °C prior to cytokine analyses.

A specific IFN-γ cytokine ELISA was conducted as a representation of the key cluster of differentiation (CD)4 T cell subset T-helper (Th)-1 response signature cytokines, as previously described [37] for Studies A and B. Interleukin (IL)-10, IL-4 and IL-17A were assessed as representative CD4 T cell signature cytokines in quantifiable sandwich ELISAs as previously described [37] with the exception of using a quantifiable recombinant ovine IL-17A set of standards in the Kingfisher Biotech ovine IL-17A ELISA (#DIY0925V, Kingfisher Biotech, St. Paul, MN, USA) for culture supernatant samples for Study B only.

### 2.10. Assessment of Humoral Responses to ORFV

The sera were analysed for the presence of ORFV-specific immunoglobulin (Ig)-G using an in-house ELISA previously adapted [38] using the previously published generic protocol [39]. In brief, high-binding serological plates (Griener-Bio One #M129B, Glasgow, UK) were coated with 50 µL/well of cell-culture derived Orf virus antigen or foetal lamb muscle cell culture supernatant at a 1:500 dilution in 0.1 M carbonate buffer, pH 9.6 overnight at 4 °C. Plates were washed six times in PBS with 0.05% Tween 20 (Merck, #P4474, #P1379) (wash buffer) prior to blocking in 4% Infasoy (Cow & Gate, Wiltshire, UK, #COW495N) diluted in PBS for 1 h at RT. Plates were washed twice, and 50 µL/well of diluted serum test samples (at 1:200 and 1:500 in 2% Infasoy/PBS) were added in duplicate for each of the positive and negative antigens alongside known positive and negative control sera for 1.5 h at RT. Plates were washed six times, and the 50 µL/well of detection antibody donkey anti-sheep IgG-horseradish peroxidase (Binding Site, Thermo Fisher Scientific, Cat no: 258122B), at a 1:2000 dilution for 1 h at RT. Plates were washed six times, and then 50 µL/well of tetramethylbenzidine ELISA substrate was added for 5 min for development prior to reaction termination with 50 µL/well of 0.2 M hydrochloric acid. Plates were read by spectrophotometer (Dynex MRX II, Dynex Technologies, Chantilly, VA, USA) at 450 nm using the Revelation™ software (Version 4.25). Corrected sample optical density (OD) values were calculated by the subtraction of the OD value of test sera bound to negative FLM antigen.

### 2.11. Statistical Analyses

Cytokine responses to each stimulation condition measured over bleeds for each vaccine and unvaccinated control group were modelled using linear mixed models fitted by restricted maximum likelihood to ln (x + 1) transformed data. Analogous models were fitted to log-transformed cellular immune responses and humoral responses to MOMP and the vaccine backbone ORFV. All models included treatment group, bleed and their interaction as fixed effects and animal ID as a random effect. The statistical significance of model coefficients was based on conditional *F*-tests. Post hoc pairwise comparisons were based on *t*-tests from estimated marginal means. Whenever multiple statistical comparisons over treatment groups or bleeds were performed, the resulting *p*-values were adjusted to control for a false discovery rate using the Benjamini–Hochberg method [40].

Statistical modelling and associated comparisons were conducted on the R system for statistical computing version 4.4.1 [41]. Statistical significance was concluded at the ordinary 5% level.

## 3. Results

### 3.1. Confirmation of ompA Gene mRNA/MOMP Protein from Vaccine Constructs In Vitro

The sequencing of the products of specific PCRs from FLS cells infected with the mORFV-*ompA* vaccine revealed that the recombinant virus was transcribing the RNA, and this was not derived from *g*DNA (Appendix A). The expression of MOMP from the mMVV-*ompA* vaccine was determined by transduction of cultured CRFK cells and immunoblot analysis of cell lysates. Immunoblotting showed the presence of the expected 40 kDa MOMP-FLAG protein (Appendix A).

### 3.2. Screening of Antigen Reactivity to MOMP in Sheep Cohort for Study A

#### 3.2.1. Antibody Responses to MOMP

Serum samples from day −10 (Figure 4) were assessed for the cohort of sheep identified for the comparative immunogenicity Study A. Sheep were selected for the *ompA*-vaccine groups that met the criteria of IgG-MOMP responses under the designated kit cut-off of positivity under 50% S/P ratio (Appendix A). There was one animal, #30, that had a baseline value of 72.38 and was deemed to be positive; thus, that animal was kept in the final cohort of 12 within the unvaccinated control group. The three other sheep in the unvaccinated group had baseline values of 34.61, 16.48 and 23.74.

#### 3.2.2. Screening of Sheep Cohort for Assignment of Vaccine and Control Groups by Evaluation of Cellular Baseline Responses to Concanavalin A and Recall Responses to Chlamydial Antigens

PBMC responses of LSAs to baseline bleeds at days −94 and −48 were used to identify suitable sheep for Study A. Sheep were selected on the basis of high responses to ConA, low responses to COMC and EBs and negligible responses to media alone (Appendix A). On the basis of the baseline serological MOMP and cellular IFN-γ response data, a suitable cohort of sheep was identified and assigned into experimental groups, as described in Section 2.6.

### 3.3. Humoral and Cellular Antigen-Driven Responses to MOMP Following Immunisation with ompA Vaccines

#### 3.3.1. Anti-MOMP IgG Responses

There were negligible anti-MOMP responses following PI (bleed 3, Appendix A) for sheep in the mORFV-*ompA* group and the mMVV-*ompA* group. However, at bleed 6 following the BI, three of the four mORFV-*ompA* sheep had strong antibody titres of 157.96, 136.54 and 248.76% S/P ratios. In the mMVV-*ompA* group, antibodies were elevated in two of the four sheep but only deemed to be borderline positive in a single animal with an S/P ratio of 57.11%, just below the 60% S/P cut-off for positivity. The antibody level (expressed as S/P) of the post-abortion *C. abortus*-positive sheep was consistently over 400% (496.32, 498.58 and 439%).

#### 3.3.2. Cellular IFN-γ Responses Following Vaccine-Delivered *ompA*

The responses across the groups were generally high for responses to the mitogen ConA and negligible to media alone (Appendix A). Chlamydial antigen responses, where present, were generally stronger to the COMC antigen rather than the whole EB antigen throughout Study A. Following PI, two of the four mORFV-*ompA* sheep had elevated IFN-γ responses to COMC beyond baseline values, whereas only one of the mMVV-*ompA* sheep (#27) had such elevated responses. Consistently, two sheep in the mORFV-*ompA* had elevated COMC responses, but just a single mMVV-*ompA* animal showed elevated responses inconsistently over the course of the study from PI. Following the booster immunisation (BI), all four mORFV-*ompA* sheep had elevated COMC-driven IFN-γ (Appendix A) at bleed 7, but just a single mMVV-*ompA* sheep (#26) showed elevated responses over baseline levels.

Following analysis of the IgG-MOMP (Appendix A) and COMC-specific IFN-γ responses (Appendix A), the mORFV-*ompA* group had a higher proportion of sheep (three of four; 75%) responding to the vaccine antigen post-booster immunisation (antibody and cellular responses in combination) than the mMVV-*ompA* group (only one of four sheep responded with either elevated antibody or cellular IFN-γ). The magnitude of these humoral and cellular antigen-driven responses was also greater in the mORFV-*ompA* group (Appendix A). Given these results, it was decided to progress with the mORFV-*ompA* vaccine in a new cohort of sheep. Given that both live and inactivated parapox virus preparations are known to stimulate vaccine-induced responses, we decided to assess β-propiolactone inactivated mORFV-*ompA* alongside the live vector in the comprehensive follow-up immunogenicity Study B (Figure 5).

### 3.4. Evaluation of mORFV-ompA Inactivation

The absence of CPE following three rounds of successive sub-culture of the inactivated virus in FLS cells was noted. Analyses of RNA from these cultured cells from three sub-cultures revealed specific mRNA, in the absence of *g*DNA, of transcripts encoding *ompA* from sub-culture round one only. This suggests that no viable mORFV-*ompA* was present following two rounds of β-propiolactone inactivation. It was not possible to titrate the inactivated mORFV-*ompA*. The live mORFV-*ompA* was titrated as previously described (in Section 2.2 above).

### 3.5. Comprehensive Immunogenicity Evaluation of Live and Inactivated mORFV-ompA In Vivo in Sheep

#### 3.5.1. Screening of Sheep Cohort for the Comprehensive Immunogenicity Study B Comparing Live and Inactivated mORFV-*ompA*

The final three groups of 10 sheep were identified from a cohort of 50 sheep on the basis of negative test results to the pestivirus screen (MRI Virus Surveillance Unit), negative MOMP-IgG responses (Figure 5, bleeds 1 and 2) and specific PBMC IFN-γ responses across bleeds 1 and 2 (high to ConA, low/negligible to COMC, EB and medium alone, Appendix A).

#### 3.5.2. Humoral Responses to MOMP in ORFV-*ompA* Immunisation Study B

The mean antibody (whole IgG) responses presented as S/P ratio for each vaccine group and control group are presented in Figure 5 (raw data presented in Appendix A). There were no responses across the three groups pre-immunisation (bleeds 1 and 2), but an immediate significant increase was observed in both vaccine groups PI (*p* < 0.0001, bleed 3, Figure 5). The magnitude of the antibody response of the live vaccine was stronger than that of the inactivated vaccine, but these were not significantly different from each other (*p* = 0.0747). The response of the vaccine groups (live and inactivated mORFV-*ompA*) started to drop slightly at bleed 4. At bleed 5, following the BI, there was a strong vaccine antigen-specific IgG response to both vaccines and again, the magnitude of this response in the live group was greater, and this was maintained for the duration of the study (Figure 5, bleeds 6–8). Again, there is a significant increase in these vaccine responses (bleed 5 vs. bleeds 6–8, *p* < 0.0001) but not significantly different to each other (live vs. inactivated mORFV-*ompA*, bleeds 6–8 *p* = 0.5289).

#### 3.5.3. Cellular Immune Responses

The mean cellular cytokine responses of IFN-γ, IL-17A, IL-10 and IL-4 for each vaccine and unvaccinated control group data are collated according to antigen and recall assay stimulation condition (medium, COMC, EB and ConA) in Appendix A with only the medium and ConA presented in bar graphs for each cytokine in separate Appendix A. At the pre-immunisation sample points (bleeds 1 and 2), responses were negligible for both the medium alone and the chlamydial antigen whole EBs. Strong and consistent ConA responses were observed across groups for IFN-γ, IL-17A, IL-10, and IL-4. However, an isolated IL-4 response to the medium alone at sample point bleed 2 was noted in the live mORV-*ompA* group (Appendix A). This showed that the cohort of sheep in each group had appropriate baseline responses to the vaccine antigen and capability to respond strongly to the T cell mitogen ConA (Appendix A).

Across Study B, responses to medium alone remained at baseline levels broadly for all sample points for each cytokine (Appendix A). The polyclonal T cell responses to ConA served as a positive control in the lymphocyte stimulation assays and have shown consistent cytokine production between the vaccine and unvaccinated groups (Appendix A) but marginal variable responses between groups over time for IFN-γ (*p* = 0.0464) but not the other cytokines (IL-17A, *p* = 0.2019; IL-10 = 0.7726 and IL-4, *p* = 0.1819).

#### 3.5.4. Cellular Immune Responses to the MOMP Antigen in Sheep Following Immunisation with mORFV-*ompA*

The mean cellular IFN-γ, IL-17A, IL-10 and IL-4 responses to the chlamydial antigens COMC and whole EB both showed vaccine-induced responses, but there was a level of background for EB IFN-γ responses prior to PI. No EB responses above baseline across the groups and time were observed for IL-17A, IL-10 and IL-4. For these reasons, the EB response has been presented as raw data (Appendix A).

The mean cellular IFN-γ responses to the chlamydial antigen COMC for each vaccine group and control group are presented in Figure 6A. There was a difference between the experimental groups (*p* < 0.0001) in the responses observed over time (*p* < 0.0001) and the divergence of responses to vaccinations over time over the unvaccinated control group (*p* < 0.0001). No COMC-specific responses were detected across the groups pre-immunisation (Figure 6A–D, bleeds 1 and 2). Within two weeks of PI, there was a significant upregulation in COMC-IFN-γ response observed in both vaccine groups over the unvaccinated control group (Figure 6A, bleed 3, live and inactivated group vs. unvaccinated control group *p* < 0.001). These elevated vaccine–antigen recall responses were broadly maintained in the live group, but responses were more variable in the inactivated group (bleeds 3, 4 and 5, live vs. inactivated mORFV-*ompA* groups). In response to BI, there was a gradual but significant increase in the COMC-IFN-γ response in the vaccine groups, peaking at bleed 7 (approx. 4 weeks BI). The live mORFV-*ompA* group had a stronger mean response post-BI than the inactivated ORFV-*ompA* group, but the levels were not significantly different from each other. Over the entire study, there was no difference in the COMC-IFN-γ response between the live and inactivated groups (*p* = 0.9391).

The mean cellular IL-17A response to COMC from all groups is presented in Figure 6B. There was a significant difference between the groups (*p* < 0.0039) overall, that is, in response to the vaccinations over time (*p* < 0.0001). Again, there was divergence over time between the vaccine groups and the unvaccinated group (*p* = 0.0002). The responses pre-immunisation were negligible (Figure 6B, bleeds 1 and 2), but following PI, IL-17A is primed in both live and inactivated mORFV-*ompA* groups (Bleeds 1, 2 vs. 3 *p* = 0.0002) to a similar level as the IFN-γ response (Bleed 3 Figure 6A,B). A consistent pattern of the PI IL-17A responses was observed, similar to that between PI and BI IFN-γ responses in both vaccine groups (Bleeds 3–5, Figure 6A,B). Following BI, the IL-17A response in the live group was maintained and slightly elevated in the inactivated group, but no significant differences between vaccine groups over time were determined (*p* = 0.332).

Analysis of the mean regulatory IL-10 group responses (Figure 6C) and the mean anti-inflammatory IL-4 responses (Figure 6D) to the COMC antigen revealed no elevation of responses to either immunisation in each live or inactivated vaccine group above those observed prior to immunisation (IL-10 between-group comparison, Figure 6C, *p* = 0.501; IL-4 between-group comparison, Figure 6D, *p* = 0.1299). Incidentally, the magnitude of these responses was at the sensitivity limit of both IL-10 and IL-4 ELISAs.

#### 3.5.5. Humoral Responses to the Vaccine Backbone, ORFV

The mean specific antibody (IgG) responses to ORFV for each group are presented in Figure 7, and the raw data is included in Appendix A. Analyses of the IgG responses to ORFV revealed a high degree of variability between all three groups (both vaccine groups and unvaccinated control, *p* = 0.0005) and between the groups for the duration of the study (*p* < 0.0001). Analysis of the pre-immunisation bleeds 1 and 2 revealed that all sheep across the three groups had antibodies to ORFV, which suggests prior exposure to the virus. It seems that the unvaccinated group had slightly lower responses in these pre-immunisation sample points than the live and inactivated vaccine groups. This response was around an OD of 1.5 and remained fairly constant over the duration of the study, apart from a transient drop at bleed 6 also found across the vaccine groups. For the live and inactivated groups, the responses started at around an OD of 1.9 to 2.1 and deviated by ODs of 0.5 over time. There was no consistent additive response to ORFV following immunisations (PI and BI) of both vaccine groups, suggesting that the vaccines are not stimulating responses to the live or inactivated mORFV vector backbone. Overall, there were no divergent responses between the live and inactivated vaccine group responses over the study (*p* = 0.5543) or, indeed, between the three groups together over time (*p* = 0.7743) and are quite different to the antibody response to the MOMP antigen post immunisation(s).

## 4. Discussion

The aims of this project were to develop prototype vaccines using modified virus vector platform technologies derived from ORFV and MVV to determine their potential utility in delivering a bacterial *C. abortus* protein as a transgene product and stimulate potentially protective immune responses in sheep. The MOMP protective antigen has formed the basis of many prototype chlamydial vaccines across human, farm animal and wildlife developmental vaccines. It comprises approximately 60%, by protein weight, of the outer surface of known adjuvanted experimental COMC-based vaccines [26]. The prototype mORFV-*ompA* and mMVV-*ompA* vaccines were both shown to incorporate the *ompA* gene with a demonstrable expression of MOMP in vitro in susceptible cells. A direct comparison of these vaccines in sheep showed that they both stimulated antigen-specific IgG and cellular IFN-γ in at least 50% of the immunised sheep. This is the first report of a direct comparison of modified endemic UK sheep viruses (ORFV and MVV) stimulating immune responses to foreign transgenes in vivo in sheep. The humoral responses were stronger to the ORFV-based vaccine following the BI in the primary Study A, but wider conclusions about relative vaccine platform performance should be drawn with caution given the group sizes here. This is especially the case for sheep antibody responses to the MOMP vaccine antigen used in this project since it has been previously reported that not all post-abortion sheep respond to MOMP following experimental inoculation with *C. abortus* [42,43].

The cellular IFN-γ response to immunisation with mORFV-*ompA* was again stronger than the mMVV-*ompA* response for the number of responding animals and the magnitude of this response. For mMVV-*ompA,* the cellular IFN-γ response was rather limited and highly variable. This could be a reflection of the cellular assay system used to assess the cellular immune responses and/or a reflection of differences in the capacity of each platform to stimulate responses to the vaccine antigen here. Both ORF and MVV are known to target different cells in sheep. ORF is epitheliotrophic, and MVV is known to target antigen-presenting cells such as macrophages and dendritic cells [44,45]. It could be that one or both vaccine platforms are more suited to stimulating CD8 T cell responses rather than the predominant responses measured in a PBMC restimulation assay such as CD4 T, natural killer and gamma-delta T cells. Amann et al. have shown ORFV to stimulate both CD4 and CD8 T cell responses in non-permissive hosts [46]. Further investigations are required to fully evaluate the relative capacities of these platforms for stimulating immune responses in a larger cohort of animals. This dataset demonstrates that both vaccine platforms are able to deliver a bacterial antigen that primes an antigen-driven immune response in vivo. The mMVV-*ompA* system has been shown to transduce ovine monocyte-derived dendritic cells in vitro, leading to the release of IL-1beta and the production of apoptotic bodies. Immature dendritic cells have then been capable of stimulating IFN-γ production in T cells [25].

Following on from the promising results of the mORFV-*ompA* in Study A, where there had been some evidence of prior *C. abortus* exposure within the cohort of sheep used, a wider investigation of the utility of this parapox-based vaccine in a new cohort of sheep derived from OEA-free Scottish Premium Health Scheme flocks was undertaken. Here, in a larger cohort of sheep, the capacity of both live and inactivated vaccines to stimulate MOMP-antibody and the quality of the antigen-driven CD4 T cell responses was evaluated. The vaccines stimulated a humoral response to MOMP observed PI but much stronger post-BI. The live vaccine elicited a higher observed response following immunisation, but this was not statistically significant. In terms of utility as a vaccine platform, the ability to stimulate antibody responses using inactivated preparations adds to the safety profile of this type of vaccine, especially increasing potential opportunities for deployment. Incidentally, for the control of *C. abortus* in ovine chlamydiosis (ovine enzootic abortion), antibody responses are considered to play a minor role in control in sheep but could be important in preventing re-infection [43].

The evaluation of the quality or balance of the antigen-driven cellular immune response to these vaccines was undertaken to determine the comprehensive balance of the CD4 T cell signature cytokines stimulated. The live vaccine stimulated a strong balance of IFN-γ and IL-17A following the two-dose regimen with no observable production of IL-10 or IL-4. This type of dominant T-helper (Th)-1/Th-17 biased response has been associated with host responses to chlamydial infections at various levels of disease severity across human, farm animal and wildlife hosts [37,47,48]. IFN-γ is well regarded as the key cytokine responsible for controlling chlamydial species and within a Th-1 targeted vaccine-induced response to the current live *C. abortus* vaccines [27,49].

Emerging evidence has identified that IL-17A in isolation can be partially protective in a rodent vaccine-challenge model of *C. trachomatis* on Th-1 deficient background [50]. In addition, an immunogenicity trial using an adjuvanted *C. trachomatis* chlamydial protease activity factor vaccine showed elevated IFN-γ and IL-17A in immunised pigs [51]. Together with recently published vaccine-challenge study data from our group, where we showed that the prototype COMC vaccine adjuvanted with ISA VG 70 (Seppic, Paris, France) stimulated IFN-γ production, but negligible IL-10 and IL-4 in cellular assays to COMC antigen in sheep protected from OEA [27], this suggests that this platform may be useful for the development of a next-generation vaccine to OEA.

Both the live and inactivated vaccines in this study stimulated a similar cellular immune response with subtle differences in the relative levels of IFN-γ and IL-17A post-BI. This is a valuable and important first step towards mORFV-based vaccine development that provides information on immune responses in the natural host. Further investigations, such as full vaccine-challenge studies in pregnant sheep, are required to determine what the relative alterations in these cytokine levels could mean for potential protection against OEA. If efficacious, the use of an inactivated ORFV-*ompA* vaccine has the potential to increase the safety profile of this virus vector construct-based type of vaccine.

One of the major features of using ORFV-based vaccines is that previous exposure does not prevent re-infection with the virus. Therefore, it is believed that any ORFV-specific responses induced could be less problematic than other viral vaccine vectors where the immune response to the vector could inhibit it from expressing the ‘vaccine/foreign antigen’ [13,14]. In this study, the vaccine-induced humoral and cellular immune responses were conferred in sheep despite prior exposure to ORFV and systemic IgG-ORFV. Valuable immunogenicity data for a common bacterial antigen is presented here, which adds to the portfolio of information available on parapox vaccine platforms for the magnitude and quality of vaccine-induced immune responses. This contributes valuable data and knowledge to a dossier of information that, in turn, contributes to the Platform Technology Master File. This would accelerate the route to market for vaccines based on known platform technologies for future next-generation vaccine registration through the European Medicines Agency [52].

The utility of poxvirus-based vaccines is already widely adopted in commercial vaccines for pets and food animals, including the trivalent vaccine to myxomatosis and rabbit haemorrhagic disease virus genotypes 1 and 2 [53]. Therefore, many of the potential barriers to the commercialisation of such a vaccine platform for use in veterinary species, such as scalability, formulation and purification, have already been evaluated [54,55,56].

## 5. Conclusions

A mORFV-*ompA* vaccine delivered as a live or inactivated preparation has the ability to stimulate MOMP bacterial antigen expression in sheep despite prior exposure to ORFV. The presence of vector-specific immunity (ORFV-specific IgG) did not change following immunisation and did not prevent responses to the vaccine antigen MOMP. Immunisation following a two-dose regimen stimulated both systemic IgG to MOMP and antigen-specific cellular responses dominated by IFN-γ and IL-17A production, immune response signatures shown previously to be protective against chlamydial disease in representative vaccine-challenge models. These mORFV-*ompA* prototype vaccines offer the possibility of a new generation of vaccines for bacterial pathogens in sheep. If efficacious, the inactivated vaccines offer enhanced levels of safety, providing greater opportunities for potential deployment.

## Figures and Tables

**Figure 1 vaccines-13-00631-f001:**
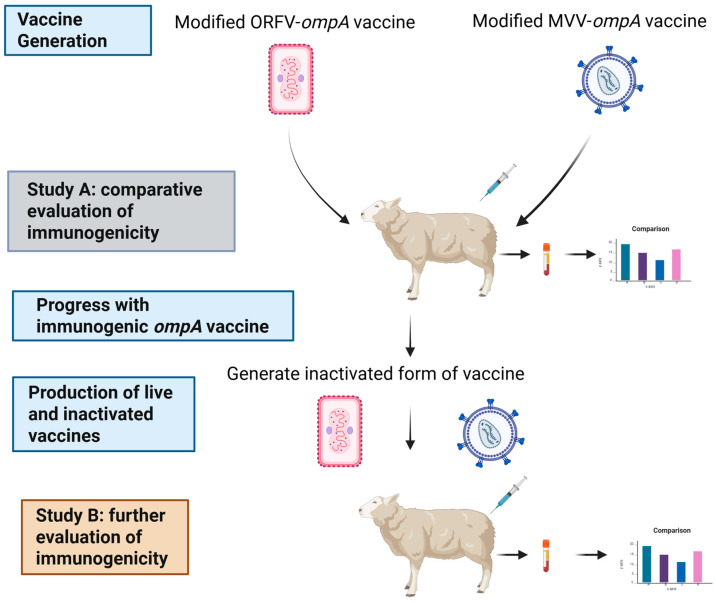
Overview of the project. The generation of the two vaccines from a modified Orf virus (mORFV) vector and a modified maedi-visna virus (mMVV) vector containing the *ompA* gene, and their assessment in sheep Study A. Here, the measurements of the major outer membrane protein (MOMP)-specific antibody and antigen-recall lymphocyte stimulation assays for IFN-γ production from culture supernatants were undertaken. The most promising vaccine candidate was identified and progressed for evaluation in a new cohort of sheep for comprehensive immunogenicity assessment of live and inactivate versions (Study B). Measurements of MOMP antibody, anti-vector antibody and cellular supernatants for IFN-γ, interleukin (IL)-17A, IL-10 and IL-4 production were performed. Created in BioRender, https://BioRender.com/76jx35t (accessed on 19 May 2025).

**Figure 2 vaccines-13-00631-f002:**
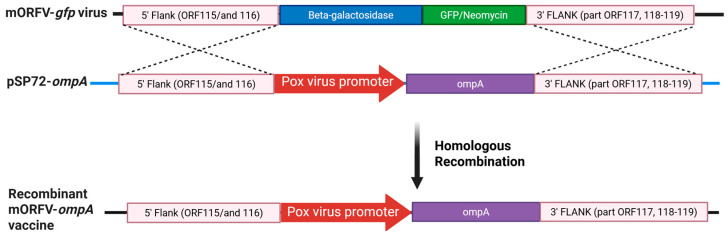
Modified ORFV-*ompA* vaccine generation. Described in detail in Section 2.2. Created in BioRender, https://BioRender.com/j9igzmk (accessed on 19 May 2025).

**Figure 3 vaccines-13-00631-f003:**
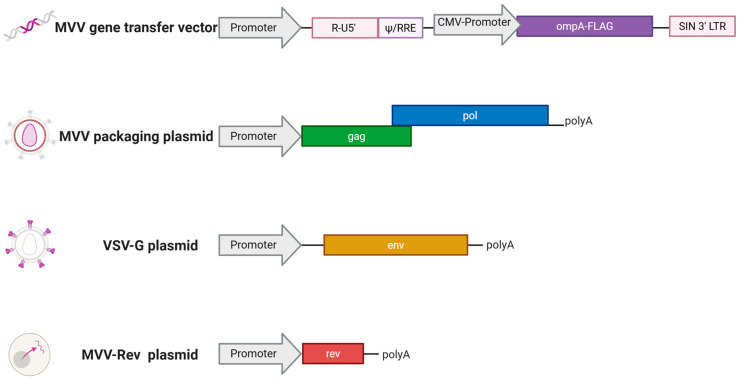
Modified MVV-*ompA* components used for vaccine generation. The recombinant virus vaccine production is described in detail in Section 2.3. Created in BioRender, https://BioRender.com/k0tubqq (accessed on 19 May 2025).

**Figure 4 vaccines-13-00631-f004:**
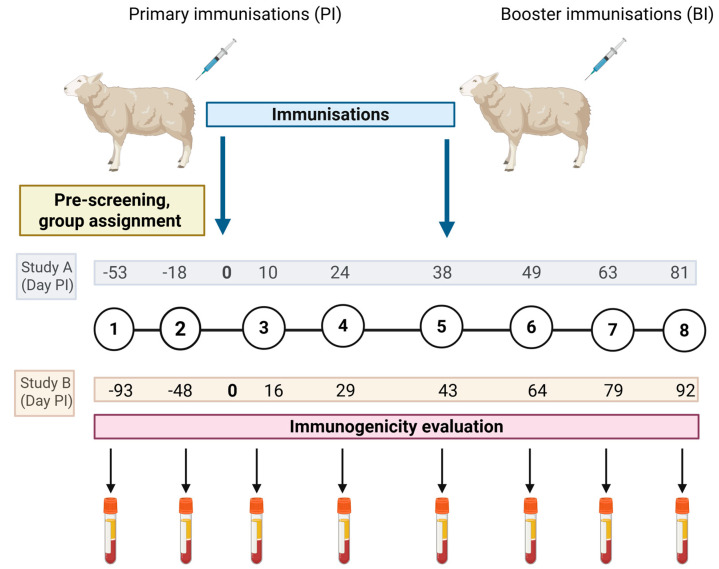
Overview of Immunogenicity Studies A and B. This is an overview of the format of each immunogenicity study (A and B), comprising eight blood sampling points represented in the black numbers within the black inter-linked circles and two immunisations represented with the blue arrows. In these studies, two separate blood samples were taken (for serum and peripheral blood mononuclear cell preparation for lymphocyte assay set up as described in Section 2.6) to screen the cohort of sheep (*n* = 30 for Study A, *n* = 50 for Study B) for assignment of experimental groups (following methodology described in Section 2.6). A primary immunisation (PI) was delivered, followed by three further blood samples. A booster immunisation (BI) was delivered prior to three final blood samples. Specifics relating to Immunogenicity Study A: a study comparing the immunogenicity of mORFV-*ompA* and mMVV-*ompA* immunisations in sheep alongside an unvaccinated control group. The eight blood samples taken relative to PI are stated in days in the grey box. Specifics relating to Immunogenicity Study B: a comparison of live and inactivated *ompA*-vaccines in immunised sheep alongside a group of unvaccinated controls. The eight blood samples taken relative to PI are indicated in days in the beige box. Created in BioRender, https://BioRender.com/mz4dpae (accessed on 19 May 2025).

**Figure 5 vaccines-13-00631-f005:**
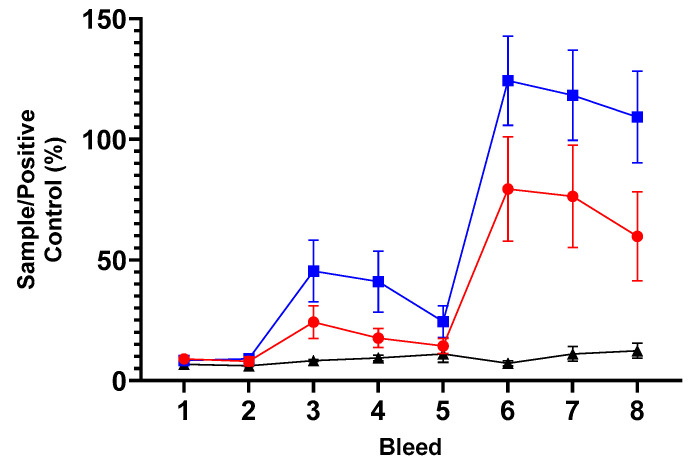
Serological responses to MOMP. The arithmetic mean of antibody immunoglobulin (Ig) G response to MOMP antigen presented as a percentage of the Sample/Positive Assay Control (% S/P). The analysis of sera by ELISA following the manufacturer’s guidance outlined in Section 2.8 from sheep grouped by immunisation groups or unvaccinated control groups for each of the eight bleed sample points is outlined in Figure 4 (using raw data from Appendix A). Briefly, bleeds 1 and 2 were prior to the primary immunisation; booster immunisation occurred immediately after bleed 5 and, therefore, bleeds 6–8 constituted post-booster sample points. The blue line with filled square symbols represents the live ORFV-*ompA* group, the red line with filled circle symbols represents the inactivated ORFV-*ompA* group, and the black line with filled triangle symbols represents the unvaccinated control group. The error bars for each dataset are derived from the standard error of the mean (standard deviation/square root of N, SEM).

**Figure 6 vaccines-13-00631-f006:**
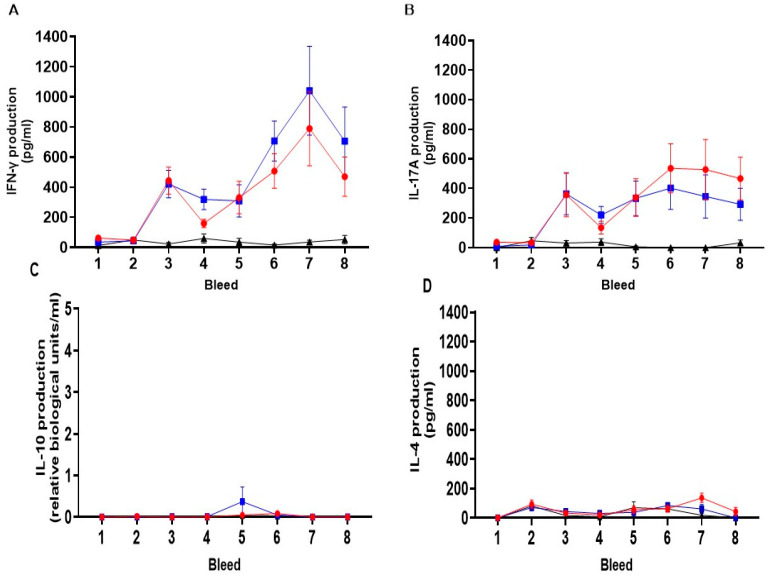
Cellular IFN-γ (**A**), IL-17A (**B**), IL-10 (**C**) and IL-4 (**D**) responses to chlamydial outer membrane complex from peripheral blood mononuclear cells. The arithmetic mean of cytokine (IFN-γ, IL-17A, IL-10 and IL-4) responses from the analysis of culture supernatants derived from PBMC lymphocyte stimulation assays cultured for 96 h as stated in Section 2.6 and analysed by specific ELISAs outlined in Section 2.9. The data was grouped and presented by sheep immunisation and unvaccinated control groups for each of the eight bleed sample points outlined in Figure 4 in separate graphs for each cytokine (using raw data from Appendix A). The units of measurement for IFN-γ, IL-17A and IL-4 are by protein weight picograms/mL, and IL-10 is relative biological units/mL. The blue line with filled square symbols represents the live ORFV-*ompA* group, the red line with filled circle symbols represents the inactivated ORFV-*ompA* group, and the black line with filled triangle symbols represents the unvaccinated control group. The error bars for each dataset are derived from the standard error of the mean (standard deviation/square root of N, SEM).

**Figure 7 vaccines-13-00631-f007:**
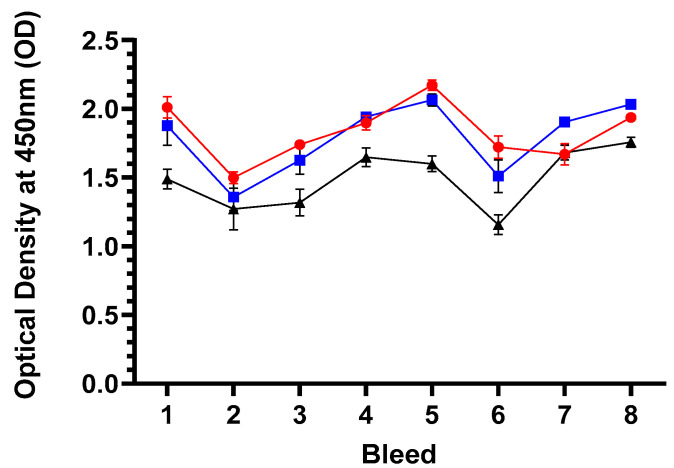
Serological responses to Orf virus. The figure shows the arithmetic mean of antibody immunoglobulin (Ig) G response to Orf virus presented as optical density values at 450 nm. The sera were analysed by ELISA following the protocol in Section 2.10. Sheep were grouped by immunisation group or unvaccinated control group for each of the eight serological sample points outlined in Figure 4 (using raw data from Appendix A). Briefly, bleeds 1 and 2 were prior to the primary immunisation; booster immunisation occurred immediately after bleed 5 and, therefore, bleeds 6–8 constituted post-booster sample points. The blue line with filled square symbols represents the live mORFV-*ompA* group, the red line with filled circle symbols represents the inactivated mORFV-*ompA* group, and the black line with filled triangle symbols represents the unvaccinated control group. The error bars for each dataset are derived from the standard error of the mean (standard deviation/square root of N, SEM).

## Data Availability

Data are contained within the article or Appendix A. The original contributions presented in the study are included in the article/Appendix A, and further inquiries can be directed to the corresponding author/s.

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
