# Peer review of "Evaluation of Immunogenicity of an Orf Virus Vector-Based Vaccine Delivery Platform in Sheep"

_vaccines, 2025, doi:10.3390/vaccines13060631_

Round 1
Reviewer 1 Report
Comments and Suggestions for Authors
This is a very well-presented manuscript of a scientifically designed and appropriately executed study. In my judgment this study and the results presented add a significant value to the field of this research area.
Minor comments/suggestions:
Title could be "Evaluation of immunogenicity of an Orf virus-based vaccine delivery platform in sheep".
Methods section 2.2 - Lines 147 and 149 - "using standard techniques" - providing/adding references to these techniques would be more appropriate.
Reviewer 2 Report
Comments and Suggestions for Authors
This manuscript presents a detailed immunogenicity assessment of an Orf virus (ORFV) and maedi-visna virus (MVV)-based vaccine vector expressing Chlamydia abortus MOMP (ompA) antigen in sheep. It includes the development and in vivo comparison of two viral platforms, providing valuable data for bacterial vaccine vector research, an area still underdeveloped compared to viral targets. The study is technically well executed, but the manuscript would benefit from a more concise introduction, a stronger contextualized discussion, and clearer graphical presentation of some data. The use of an endemic parapoxvirus vector for delivery of bacterial antigens in sheep is a novel contribution, particularly in the context of veterinary vaccine platforms. This work fills a critical gap in the development of bacterial vaccines using viral vectors and offers a potentially scalable strategy for veterinary immunization. While scientifically strong, the manuscript would benefit from editing for clarity, particularly in the introduction and figure legends. Rigorous experimental design, use of appropriate controls, validated assays, and robust statistical modeling support scientific integrity.This study is of interest to researchers in veterinary vaccinology, immunology, and viral vector development, particularly those working on bacterial disease control.
The introduction is overly long and saturated with references (31), many of which are more appropriate for the Discussion section. Key concepts such as the gap in bacterial vector vaccine technology and the rationale for using ORFV should be emphasized earlier. Citations should focus on foundational studies and recent breakthroughs only.
The two-phase study design comparing mORFV and mMVV vectors, followed by a detailed immunogenicity comparison of live versus inactivated ORFV vaccines in sheep, is logical, robust, and well justified. Use of OEA-free flocks and pre-screening enhances internal validity.
The methodology is detailed to a high technical standard, with sufficient information for replication. However, the manuscript would benefit from a summarizing schematic of both vaccine preparation processes. The inclusion of homologous recombination steps, plasmid construction, and cytokine profiling are exemplary.
While the quantitative data are comprehensive, the presentation is at times dense and difficult to navigate. Key figures (e.g., Figures 5–7) are informative but would be clearer with more consistent Y-axis scaling, explicit statistical annotations, and simplified legends. Supplementary tables are essential but underutilized in text interpretation.
The conclusions regarding the superior immunogenicity of the mORFV vector and the potential for both live and inactivated platforms are well supported by both humoral and cellular data. However, discussion of the translation of immunogenicity into protective efficacy (e.g., challenge studies) is lacking and should be acknowledged as a limitation.
Figures require clearer legends, indication of statistical significance (e.g., p-values, asterisks), and better labeling. The use of color is effective but should comply with colorblind-friendly practices. A schematic summary figure of the experimental workflow would help orient readers.
Recommendations for Authors
Introduction Revision: Reduce to 5–6 key paragraphs focused on rationale, novelty, and research question. Relocate mechanistic and historical details to Discussion where they support interpretation.
Discussion Expansion: Integrate comparative data from other vector systems in sheep and discuss immune memory, potential protective correlates, and applicability to other bacterial pathogens. Explicitly identify gaps and propose follow-up challenge studies to validate protection.
Figures and supplementary material: Improve clarity of axis labels and legends. Include summary figure of experimental timeline and construct design. Indicate statistical comparisons directly on graphs.
Reference: Remove all self-citations. We produce science to be referenced by other researchers and not to self-reference ourselves. This is a predatory practice that this reviewer fights to abolish from the scientific community. Because self-citation makes it clear that the topic is and was studied only by the authors of the manuscript, which is not an absolute truth and/or a scientific truth. The article is excellent, you authors do not need selfcitation!
Few comments are in the attached manuscript.
Best regards
